# HPV Type Distribution in Benign, High-Grade Squamous Intraepithelial Lesions and Squamous Cell Cancers of the Anus by HIV Status

**DOI:** 10.3390/cancers15030660

**Published:** 2023-01-20

**Authors:** Sona Chowdhury, Teresa M. Darragh, J. Michael Berry-Lawhorn, Maria G. Isaguliants, Maxim S. Vonsky, Joan F. Hilton, Ann A. Lazar, Joel M. Palefsky

**Affiliations:** 1Division of Infectious Diseases, Department of Medicine, University of California, San Francisco, CA 94143, USA; 2Department of Pathology, University of California, San Francisco, CA 94143, USA; 3Department of Medicine, University of California, San Francisco, CA 94143, USA; 4Institute of Microbiology and Virology, Riga Stradins University, LV-1007 Riga, Latvia; 5Department of Microbiology, Tumor and Cell Biology, Karolinska Institutet, SE-171 77 Stockholm, Sweden; 6D.I. Mendeleyev Institute for Metrology, 190005 St. Petersburg, Russia; 7Almazov National Medical Research Center, 197341 St. Petersburg, Russia; 8Department of Epidemiology and Biostatistics, University of California, San Francisco, CA 94143, USA; 9Division of Oral Epidemiology, University of California, San Francisco, CA 94143, USA

**Keywords:** HPV genotypes, HPV 16, HSIL, anal cancer, PLWH, HIV-negative, HPV screening

## Abstract

**Simple Summary:**

Human papillomavirus (HPV)-associated cancers, such as anal cancer, are a major risk among people living with HIV (PLWH). High-risk (HR) HPVs are the causal agent of anal precancer and cancer. However, there is a paucity of data regarding the HPV genotypes that cause especially anal cancers and pre-cancers in PLWH. In this study we characterize the specific HPV types in anal tissues, including benign, pre-cancer, and cancer samples from PLWH, and compare them to similar samples from HIV-negative individuals. The results from our study suggest that a broader range of HPV types may play a role in anal cancer in PLWH than in HIV-negative individuals. Proposed screening approaches that include HPV testing might need to differ by HIV status, with extended HPV genotyping included for PLWH.

**Abstract:**

The incidence of anal cancer is increasing, especially in high-risk groups, such as PLWH. HPV 16, a high-risk (HR) HPV genotype, is the most common genotype in anal high-grade squamous intraepithelial lesions (HSIL) and squamous cell carcinoma (SCC) in the general population. However, few studies have described the distribution of HR HPV genotypes other than HPV 16 in the anus of PLWH. HPV genotyping was performed by DNA amplification followed by dot-blot hybridization to identify the HR and low-risk (LR) genotypes in benign anal lesions (n = 34), HSIL (n = 30), and SCC (n = 51) of PLWH and HIV-negative individuals. HPV 16 was the most prominent HR HPV identified, but it was less common in HSIL and SCC from PLWH compared with HIV-negative individuals, and other non-HPV 16 HR HPV (non-16 HR HPV) types were more prevalent in samples from PLWH. A higher proportion of clinically normal tissues from PLWH were positive for one or more HPV genotypes. Multiple HPV infection was a hallmark feature for all tissues (benign, HSIL, SCC) of PLWH. These results indicate that the development of anal screening approaches based on HPV DNA testing need to include non-16 HR HPVs along with HPV 16, especially for PLWH. Along with anal cytology, these updated screening approaches may help to identify and prevent anal disease progression in PLWH.

## 1. Introduction

Human papillomaviruses (HPVs) comprise a diverse group of DNA viruses that have different epithelial tropisms. HPV primarily infects squamous epithelia, including keratinized and non-keratinized skin and mucosa of the anogenital region and oropharynx; some glandular epithelia can also be infected with HPV—e.g., endocervical mucosa [1]. Most HPV infections are transitory, but persistent infections, particularly with oncogenic (high-risk) HPV types may result in anogenital precancer and cancer. HPV-associated cancers include cervical (98% positive for HPV), anal (88% positive for HPV), vaginal (74% positive for HPV), penile (50.8% positive for HPV), vulvar (18–70% positive for HPV), and oropharyngeal cancers (51.8% to 71% positive for HPV) [2,3,4,5].

The incidence of anal carcinoma in both men and women in the United States has been increasing over the years. The American Cancer Society estimated that in 2022, there would be 9440 new cases (3150 in men and 6290 in women) and 1670 deaths (930 in women and 740 in men) [6]. This increase could be attributed to changes in sexual behavior, the contribution of human immunodeficiency virus (HIV)-related immunosuppression, and, to a lesser extent, tobacco smoking [7].

Several studies have shown that the incidence of anal cancer is higher in women in comparison with men in the general population [8]. Among women, those living with HIV (WLWH) and/or lower genital tract cancer or pre-cancer (LGTN) at a site other than the anus are at increased risk for anal HPV infection and anal cancer [9,10]. Among men, anal cancer incidence is highest among men who have sex with men (MSM), particularly those living with HIV [11].

There are more than 150 different HPV genotypes that infect humans, of which approximately 40 HPV genotypes infect the anogenital tract and are sexually transmitted [12]. The HPV genotypes that infect the genitals and surrounding area can cause a range of anal lesions from condyloma acuminatum and other forms of low-grade squamous intraepithelial lesions (LSIL), high-grade squamous intraepithelial lesions (HSIL), and squamous cell carcinoma (SCC). The HPV genotypes have been classified into two risk groups based on their oncogenic potential: low-risk (LR) types, which are found mainly in external genital warts and high-risk (HR) types, which are frequently associated with HSIL and SCC. However, because of confounding data regarding the carcinogenic potential of individual HPV genotypes, there is lack of consensus regarding classification of some genotypes into HR and LR groups. For instance, the number of putative HR types varies from 13 to 19, and only 11 types (16, 18, 31, 33, 35, 39, 45, 51, 52, 56, and 58) are consistently classified as HR. According to the epidemiological classification suggested by Muñoz et al., these 11 and 4 additional HPV types should be considered as carcinogenic or HR types (59, 68, 73, and 82), and HPV types (26, 53, and 66) should be considered as probable carcinogenic [13,14].

HPV 16 is the most common oncogenic type found in cervical and anal cancer [15,16]. In people living with HIV (PLWH), a high prevalence of diverse oncogenic HR HPV infections has been reported [17,18]. In the cervix, HPV genotype distribution has been extensively studied in both HIV-negative and WLWH. It has been found that HIV-induced immunodeficiency differentially affects individual HPV genotypes; non-HPV 16 HR types play a greater role in cervical carcinogenesis compared to HPV 16 in WLWH. However, there is a paucity of data regarding the HPV genotypes associated with anal cancer, especially in PLWH, due to the rarity of anal cancer and fewer studies in this field. Given that anal cancer is the leading non-AIDS-defining malignancy in PLWH, it is important to understand the HPV type distribution in this high-risk group [19,20].

The results from the recent ANCHOR trial have unequivocally shown that treatment of anal HSIL in PLWH can reduce the incidence of anal cancer [21]. Anal cancer screening approaches will most likely include anal cytology with HPV DNA co-testing in anal swabs; thus, an understanding of the HPV types that are relevant for anal carcinogenesis is critical. These data have also important implications for shaping HPV vaccine strategy and policies.

The goal of this study is to describe the HR and LR HPV genotypes found in benign anal lesions, HSIL and SCC, and to describe the differences in the distribution of HPV genotypes in these lesions between PLWH and HIV-negative individuals.

## 2. Materials and Methods

### 2.1. Tissue Collection

Archived formalin-fixed paraffin-embedded (FFPE) anal tissues (HSIL and SCC) were retrieved from the UCSF Department of Pathology. The FFPE samples consisted of biopsies collected in outpatient clinics and surgical specimens. All blocks were de-identified for protected health information and were analyzed with institutional review board (IRB) permission. Benign anal tissue specimens were obtained from patients undergoing anal surgery for treatment of anal HSIL after obtaining their consent. Prior to electrocautery ablation of anal HSIL, the surgeon biopsied peri-anal or intra-anal areas that clinically appeared to be normal. Additional samples of clinically normal anal tissue were obtained from patients undergoing surgery for anal fistulas and hemorrhoids. These were confirmed on histopathological examination to be benign. Anal tissues were collected between 2001–2019.

Anal HSIL, SCC, and benign anal tissue from PLWH and HIV-negative individuals were included in this analysis. Information regarding age, sex, CD4 levels, HIV viral load, and anti-retroviral therapy treatment regimen(s) from a period within 6 months of the specimen collection date were obtained from the patients’ medical records.

### 2.2. Tissue Processing and HPV Genotyping

A four-micron-thick section was obtained from each FFPE block and stained with hematoxylin and eosin (H&E). H&E sections were reviewed by a pathologist to confirm the histological status of each FFPE block. To prepare DNA from the FFPE tissue, an additional 15-micron-thick section was cut from the block using a sterile technique to avoid cross-contamination.

The protocol used for HPV genotyping has been described previously [10]. Briefly, the scroll section was dissolved in CitriSolv™ (Decon Labs, Inc., King of Prussia, PA, USA), and the DNA was prepared using the RecoverAll™ Total Nucleic Acid Isolation Kit for FFPE (Thermo Fisher Scientific, Austin, TX, USA). PCR was performed using a modified pool of MY09/MY11 consensus HPV L1 primers as well as primers for amplification of the human beta-globin gene as an indicator of specimen adequacy. After 40 amplification cycles, specimens were probed with a biotin-labeled HPV L1 consensus probe mixture. A separate membrane was probed with a biotin-labeled probe to the human beta-globin gene. Specimens were also typed by hybridizing to 39 different HPV probes consisting of single and multiple HPV types: 16, 18, 31, 33,35, 39,45, 51, 52, 56, 58, 59, 66, 26/69, 30, 34, 53, 67, 68, 70, 73, 82, 85, 97, 6, 11, 54, 61, 62, 32/42, 71, 72, 81, 83, 84, 86/87, 90/106, 102/89, Mix 1 (7, 13, 40, 43, 44, 55, 74, and 91). Specimens negative for beta-globin gene amplification were excluded from analysis [22].

Specimens in which no HPV DNA was detected were assigned as “No HPV”. Specimens in which the HPV genotype could not be determined were assigned as “Type Unknown”.

### 2.3. Nomenclature

To classify HPV types into LR or HR, we used the classification as proposed by the Working Group of the World Health Organization (WHO), International Agency for Research on Cancer (IARC) [23]. HR HPV genotypes included HPV 16 and all other HR HPV genotypes in Clade 1, Group 2A and 2B, (18, 31, 33 35, 39, 45, 51, 52, 56, 58, 59, 66, 26, 69, 30, 34, 53, 67, 68, 70, 73, 82, 85, 97).

Single infection was defined as infection with one HR or LR HPV type only. Multiple infections were classified as more than one HR type (no LR types), more than one LR type (no HR types), or a combination of HR and LR types. HR HPV types other than HPV 16 was assigned as “Non-16 HR HPV”.

### 2.4. Statistical Analysis

Descriptive statistics were used to summarize the data, including frequencies and percentages for categorical data. Fisher’s exact test was used to assess differences in proportions of age, sex, HIV viral load, CD4 counts and anal disease status (benign, HSIL, SCC). Kruskal–Wallis was also used to compare age, given that the categories are ordered. For comparing the presence of HPV DNA between PLWH versus HIV individuals, stratified by anal disease, we estimated the absolute risk difference (95% exact confidence intervals (CI)).

We used generalized estimating equation log-binomial models to estimate the presence of anal disease (HSIL or SCC versus benign), with 95% confidence intervals (CI) based on the empirical covariance matrix [24]. We estimated HSIL/SCC disease presence in the overall sample, and stratified by (1) HPV 16 infection, (2) HPV 16 co-infection with HIV, and (3) HPV 16 co-infection with other HR HPV types. Models were not adjusted for covariates because the denominators of strata being compared were small. In each model, we compared infected strata to the uninfected reference level using risk ratios (95% CI) and we reported corresponding Wald 2-sided *p*-values. Two-sided *p*-values < 0.05 were considered statistically significant. SAS version 9.4 was used to generate results.

## 3. Results

### 3.1. Study Participants

A total of 115 individuals were included in this study, 34 (29.6%) woman and 81 (70.4%) men. Of these participants, 47 (41%) were HIV-negative with a median age of 57 years and 68 (59%) participants were PLWH with a median age of 52 years. Table 1a compares the age and sex of the participants stratified by HIV status. The number of male participants in the PLWH group was higher than the HIV-negative group and when participants were stratified by disease status (Table 1a,b). Age and sex were stratified by disease status for HIV-negative participants (Table 1c) and PLWH (Table 1d). Medical characteristics were also compared between disease status of PLWH (Table 1d). Of PLWH, 58 of 68 (85.3%) were on ART. The HIV viral load was undetectable in most study participants, except one untreated participant in the SCC group who had an HIV viral load greater than 10^6^ mL^−1^. CD4 counts greater than 500 cells/mm^3^ were registered in 47.6% (10/21) of participants with benign lesions, 23.8% (5/21) of participants with HSIL lesions, and 34.6% (9/26) of participants with SCC lesions.

One tissue sample from each of the 115 participants was studied, including 34 benign (<LSIL) anal lesions, 30 HSIL lesions, and 51 SCC lesions. Amplified DNA was obtained from all samples and tested for the presence of HPV DNA. Viral HPV DNA was detected in 50.0% (17/34) of benign samples, 97% (29/30) of HSIL samples, and 98% (50/51) of SCC samples irrespective of HIV status. The results of DNA testing by disease status are shown in Table 2.

#### 3.1.1. Benign Samples

HPV DNA was detected in 61.9% (13/21) of benign samples from PLWH compared to 30.8% (4/13) of samples from HIV-negative individuals, yielding an excess risk associated with HIV+ of 31.1% (95% CI, −1.4% to 63.7%). More than seven distinct HPV genotypes (HPV 16, 39, 51, 53, 73, 6, 11) were identified in benign tissue from PLWH compared with two HPV genotypes (HPV 16, 6) in samples from HIV-negative individuals (Figure 1). Multiple infections, including multiple HR types 4.8% (1/21), multiple LR types 23.8% (5/21), and a combination of HR and LR types 14.3% (3/21) were found in benign lesions from PLWH, while only single infections were found in samples from HIV-negative individuals. The proportion of multiple infections found in benign lesions from PLWH (38%; 8/21) was higher than in samples from HIV-negative individuals (0%) (*p* = 0.013, Fishers exact test) (Table 2a).

#### 3.1.2. HSIL Samples

HPV DNA was detected in 95.2% (20/21) of HSIL samples from PLWH and 100% (9/9) of samples from HIV-negative individuals, yielding a risk difference associated with HIV+ of −4.4% (95% CI, −12.2% to 4.0%). Eleven distinct HPV genotypes (HR, LR) were detected in HSIL samples from PLWH in comparison with four distinct HPV genotypes in HIV-negative individuals (Figure 1). HPV 16 with or without other types was the most prevalent HPV type in HSIL samples irrespective of HIV status: 88.9% (8/9) in HIV-negative individuals and 66.7% (14/21) in PLWH (Figure 1). However, the proportion of infection with HPV 16 alone in samples from PLWH was lower than in samples from HIV-negative individuals (42.8% (9/21) in PLWH vs. 66.7% (6/9) in HIV-negative). These results were not statistically different. In HSIL samples from PLWH, both HPV 16 and non-16 HR HPV types were found in cases of co-infections with multiple HR types and/or LR types. Non-16 HR HPV were found in combination with LR types HPV 6, 11 and 61. At the same time, in HIV-negative samples, HPV 16 was found in co-infection only with HR types 51 and 53. Only one sample from the HIV-negative group was infected with non-16 HR HPV (HPV 51) in combination with LR type (HPV 62). The proportion of multiple HPV infections found in HSIL lesions from PLWH 38% (8/21) was higher than in HIV-negative individuals 33.3% (3/9), although the difference was not statistically significant. Multiple LR HPV types, HPV 6 (14.3%), HPV 11 (9.5%), HPV 61 (4.8%), were detected in HSIL samples from PLWH, always in combination with other HPV types. The only LR type detected in the HSIL sample from HIV-negative individuals was HPV 62 (11.1%), found in combination with HPV 51. None of the HSIL samples from either PLWH or HIV-negative individuals demonstrated infection with solely LR HPVs (Table 2b).

#### 3.1.3. SCC Samples

All anal SCC samples were positive for HPV DNA except one sample belonging to the HIV-negative group (PLWH 100%, HIV- 96.0%), yielding an excess risk associated with HIV+ of 4.0% (95% CI, −3.7% to 11.7%). Fourteen distinct HPV genotypes (HR, LR) were detected in SCC samples from PLWH, while only six were found in SCC samples from HIV-negative participants (Figure 1). HPV 16 was the most prevalent HPV type in samples from both PLWH (69.2%; 18/26) and HIV-negative individuals (72.0%; 18/25). The HIV-negative group showed a higher number of SCC cases attributable to single infection with HPV 16 (68.0%; 17/25) than PLWH (42.3%; 11/26), but these results were not statistically different. In samples from PLWH, HPV 16 was found in multiple infections (27%; 7/26) with a number of HR (HPV 39, 45, 18) and LR types (HPV 6, 11, 83), whereas in the HIV-negative group only one sample was co-infected with HPV 16 and HPV 18. Non-16 HR HPV types were detected in SCC samples from PLWH, either as single infections (15.4%; 4/26) or in combination with other non-16 HR HPVs (7.7%; 2/26) or LR (3.8%; 1/26) types. In HIV-negative samples, non-16 HR HPV were only found in multiple infections in combination with other non-16 HR HPV (4.0%; 1/25) or LR (4.0%; 1/25) types. Non-16 HR HPV types were detected in 38.5% (10/26) of samples from PLWH, and 12% (3/25) of samples in the HIV-negative group. In PLWH we detected five LR types: HPV 6 (3.8%), 11 (11.5%), 83 (3.8%); and in the HIV-negative samples only two: HPV 6 (8.0%) and 11 (8.0%). Overall, few samples of PLWH and HIV-negative individuals [19.2%; 5/26) for PLWH, and (8.0%; 2/25) for HIV-negative] were positive for at least one HR type. None of the samples from PLWH and only one of the samples from the HIV-negative group contained LR HPV types only. Thus, single HR HPV infections were characteristic of SCC of PLWH while single LR HPV infections were rare. The proportion of multiple HPV infections found in SCC lesions from PLWH (38.5%; 10/26) was higher in comparison to HIV-negative individuals (16%; 4/25), although the difference was not statistically significant (Table 2c).

### 3.2. Associations of HR Types with Anal Disease

The prevalence of anal disease (HSIL/SCC) is strongly associated with HPV 16 (95.1% prevalence, compared to 42.6% among those not infected) [risk ratio (RR) = 2.23 (95% CI, 1.63 to 3.06) *p* < 0.0001] (Table 3, Model 1). We also analyzed the association of anal disease with HPV 16 and HIV infection and co-infection (Table 3, Model 2). We found that infection with HPV 16 was similarly associated with anal disease both in the absence of HIV (RR = 2.41; *p* = 0.0015) and in the presence of HIV (RR = 2.35; *p* = 0.002), and that HIV infection did not elevate the risk of anal cancer. To assess the role of non-16 HR HPV types in HPV-16-associated anal disease (Table 3, Model 3), we evaluated infection and co-infection with HPV 16 and non-16 HR HPV as risk factors (Table 3, Model 3). Infection with non-16 HR HPV types alone was associated with anal disease (RR = 3.75; *p* < 0.0001)—slightly less strongly than infection with HPV 16 alone (RR = 4.32; *p* < 0.0001). The effect of co-infection with both high-risk types was intermediate between the solo effects (RR = 4.05; *p* < 0.0001). The risk of anal disease associated with HPV 16 was much stronger when adjusted for the influence of non-16 HR HPV types (Model 3) than its unadjusted effect (Model 1).

## 4. Discussion

The aim of this study was to perform a detailed analysis of HPV genotypes found in benign, anal HSIL, and anal SCC of both PLWH and HIV-negative individuals. Understanding the distribution of different HR HPV genotypes in the spectrum of anal disease, especially in PLWH, is imperative as they are at increased risk of anal cancer.

Compared with samples collected from HIV-negative participants, four distinctive features were observed in samples collected from PLWH: (1) a lower proportion of anal HSIL and SCC that contain HPV 16 alone; (2) a higher proportion of clinically normal tissues containing HPV; (3) infection with a wider spectrum of HPV genotypes both HR and LR; and (4) an overall higher proportion of infection with multiple HPV types

The frequency of multiple infections (infection with more than one HR type, or LR types, or both) was higher in benign, HSIL, and SCC samples from PLWH compared with samples from HIV-negative participants. Higher rate of infection with more than one HR type was also found in anal cancer samples from PLWH (19.2%) versus HIV-negative anal cancers (8%). The number of LR genotypes identified in combination with HR types in samples (benign, HSIL, SCC) from PLWH was at least 5-fold to 8-fold higher than samples from HIV-negative participants. A recent study of HPV genotype distribution in MSM obtained similar results, i.e., anal infection by multiple HR HPV genotypes was higher in HIV-infected participants compared to their HIV-uninfected peers [20]. It has been hypothesized that infection with multiple HPV types may be a marker of persistent disease which could ultimately lead to disease progression [25]. As our assay does not identify amplified products at the transcriptome level, it is difficult to define the role of each HPV genotype in a multiple HPV infection. A better method to attribute a causal relationship between a particular HPV genotype and lesion formation is by the isolation of singles cells from lesions by laser capture microdissection, followed by HPV genotyping of LCM-isolated cells [26].

In this study, HPV 16 was the predominant HR HPV type found in anal HSIL and anal SCC irrespective of HIV status. Our results are similar to previous studies that have shown that HPV 16 is the most important HR type associated with anal carcinogenesis in both PLWH and HIV-negative individuals. Therefore, HPV 16 is a clear target for anal cancer prevention for both these groups [15,27]. However, we also found single infections with HPV 35 and HPV 51 in SCC samples from PLWH which are not included in the nonavalent HPV vaccine [28] and the potential to prevent HPV-related cancers in PLWH may be slightly lower than in HIV-negative individuals. Additionally, therapeutic HPV vaccines targeting PLWH may have to include a broader spectrum of HR HPV types than vaccines for HIV-negative individuals.

HPV 16 infection was found either as a single infection or in combination with other HR and LR types (multiple infection). However, the frequency of single infection with HPV 16 was higher in samples from HIV-negative individuals versus PLWH. A recent meta-analysis consisting of 95 studies showed that HPV 16 is more common in single infection in HIV-negative anal cancers and its prevalence is lower in HIV-positive anal cancers [15]. Similarly, WLWH have a lower prevalence of HPV 16 in invasive cervical cancer in comparison to HIV-negative women [29]. One of the hypotheses explaining this difference in HPV 16 frequency between these two groups (PLWH and HIV-negative) is that HPV 16 is less influenced by immunological deficiency and has better intrinsic ability to evade immune surveillance in immunocompetent individuals compared with non-16 HR HPV types [30,31]. Other than immunological deficiency, distribution of HPV types may vary depending upon other factors, including geographical distribution [32,33] and behavioral factors [34].

Consistent with the above, we found that the frequency of non-16 HR HPV types were higher in HSIL and SCC from PLWH versus HIV-negative patients. Therefore, non-16 HR HPV genotypes are more strongly affected by immunological deficiency than HPV 16, as has been reported by others [29,35].

Due to the limited number of samples in our study, we analyzed the overall prevalence of anal disease (HSIL/SCC) between PLWH and HIV-negative participants. We found that HPV 16 was strongly associated with anal disease. This association of HPV 16 with anal disease was not affected by HIV serostatus. In fact, non-16 HR HPV in absence of HPV 16 was also associated with anal disease.

The recent findings of the ANCHOR trial support the inclusion of routine screening in the detection and treatment of anal HSIL, as part of anal cancer prevention for PLWH [21]. Optimal screening programs consisting of HPV DNA testing are needed to identify PLWH at risk of developing anal cancer. Detection of low-risk HPV types alone without concurrent infection with an HR HPV type was found in only one of the cancers tested, indicating that there is no need to include these HPV types in screening tests for HSIL or SCC among either PLWH or HIV-negative populations. At the same time, our findings have shown a high representation of non-16 HR HPV types as well as an increased frequency of multiple infections in PLWH in all types of anal lesions. Although the oncogenic potential of many of the non-16 HR HPV needs to be further characterized, HPV-based screening tests focused solely on HPV 16 may not have adequate sensitivity for this group of patients. Instead, a screening test focused on a broader range of oncogenic HPV types may be needed specifically for PLWH.

Recent studies suggest that HPV latency may be more common than previously thought [36,37]. We found that more than half of the benign samples from PLWH contained HPV DNA compared to only one third of the samples from HIV-negative patients. Multiple infections in benign samples were also more common in PLWH than in HIV-negative individuals. This increased presence of HPV DNA in benign samples of PLWH could be explained due to immunosuppression that leads to increased HPV acquisition, persistence, and reduced clearance compared with immunocompetent individuals.

Our study has some limitations. The sample size is small, which limits our ability to analyze the association between the risk factors HIV and HPV genotypes at the subgroup level (HSIL, SCC) between PLWH and HIV-negative participants. Secondly, due to the cross-sectional study design, the causal relationship between the presence of a particular HPV genotype and the development of lesions cannot be established. Due to the small sample size, potentially important confounders, such as age and sex, were not adjusted for in the model and could impact the results. The results of this study need to be confirmed in a larger study. Despite these limitations, the current analyses provide valuable data regarding differences in HPV type distribution in the anus of PLWH and HIV-negative individuals.

## 5. Conclusions

In summary, we have shown that HPV 16 is the most prevalent HPV genotype in both anal HSIL and SCC irrespective of HIV serostatus. Anal HSIL and SCC in PLWH was frequently associated with infection with multiple HPV types. The increased detection of non-16 HR HPV types in PLWH underscores the need to include these genotypes in anal screening programs. Combined with anal cytology, these approaches to anal HPV type detection will need to be validated in prospective studies of PLWH and HIV-negative individuals. The detection of HPV in a high proportion of benign tissues, particularly in PLWH, is consistent with a latency state for HPV, but the clinical significance of this finding also needs further study.

## Figures and Tables

**Figure 1 cancers-15-00660-f001:**
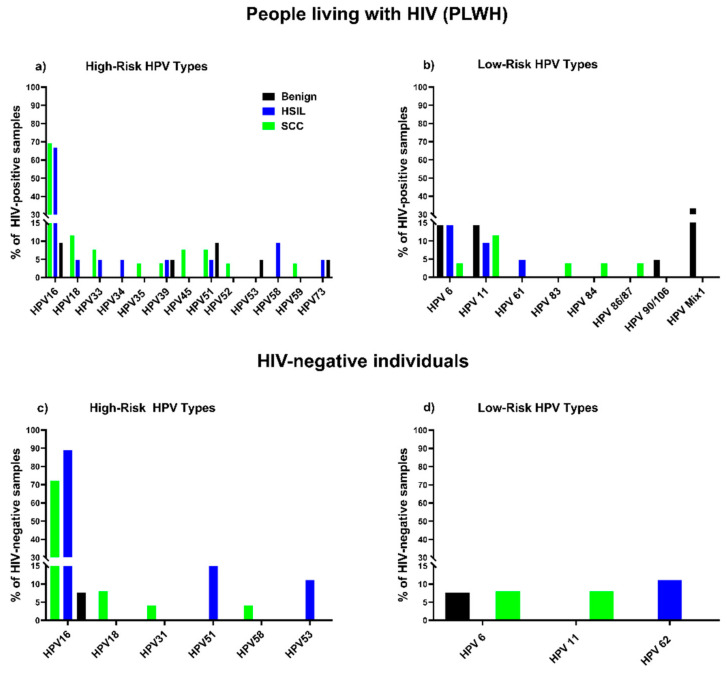
Distribution of HPV types in benign, HSIL, SCC samples: percentage of samples from PLWH containing HR types (**a**) and LR types (**b**); percentage of samples from HIV-negative individuals containing HR types (**c**) and LR types (**d**).

**Table 1 cancers-15-00660-t001:** Characteristics of HIV-negative participants and people living with HIV (PLWH) diagnosed with benign anal lesions, anal HSIL, and anal SCC.

(a) Frequency Analysis of Age and Sex of Participants Stratified by HIV Status					
Age, Years	No of Participants	PLWH		HIV-	*p*-Value
25–49	41	28 (68.3%)		13 (31.7%)	
50–69	67	38 (56.7%)		29 (43.3%)	0.065 ^#^, 0.12 ^
≥70	7	2 (28.6%)		5 (71.4%)	
**Sex**					
Male	81	61 (75.3%)		20 (24.7%)	<0.0001^
Female	34	7 (20.6%)		27 (79.4%)	
**(b) Frequency Analysis of Age and Sex of Participants Stratified by Disease Status**					
**Age, Years**	**No of Participants**	**Benign**	**HSIL**	**SCC**	***p*-Value**
25–49	41	16 (39%)	14 (34.1%)	11 (26.8%)	
50–69	67	16 (23.9%)	14 (20.9%)	37 (55.2%)	0.044 ^#^, 0.055 ^
≥70	7	2 (28.6)	2 (28.6%)	3 (42.9%)	
**Sex**					
Male	81	26 (32%)	21 (25.9%)	34 (42%)	<0.0001 ^
Female	34	8 (23.5%)	9 (26.5%)	17 (50%)	
**(c) Frequency Analysis of Age and Sex of HIV-Negative Participants**					
**Age, Years**	**Benign * (N = 13, N%)**	**HSIL (N = 9, N%)**		**SCC (N = 25, N%)**	***p*-Value**
25–49	6 (46.2%)	4 (44.4%)		3 (12%)	
50–69	5 (38.5%)	4 (44.4%)		20 (80%)	0.32 ^#^, 0.049 ^
≥70	2 (15.4 %)	1 (11.1%)		2 (8%)	
**Sex**					
Male	7 (53.8%)	3 (33.3%)		10 (40.0%)	0.68 ^
Female	6 (46.2 %)	6 (66.7%)		15 (60%)	
**(d) Frequency Analysis of Age, Sex and Medical Characteristics of PLWH**					
**Age, Years**	**Benign * (N = 21, N%)**	**HSIL (N = 21, N%)**		**SCC (N = 26, N%)**	***p*-Value**
25–49	10 (47.6%)	10 (47.6%)		8 (30.8%)	
50–69	11 (52.3%)	10 (47.6%)		17 (65.4%)	0.75 ^#^, 0.58 ^
≥70	0	1 (4.8%)		1 (3.9%)	
**Sex**					
Male	19 (90.5%)	18 (85.7 %)		24 (92.3%)	0.88 ^
Female	2 (9.5 %)	3 (14.3%)		2 (7.7%)	
**HIV Viral Load ****					
Undetectable	10 (47.6%)	13 (61.9%)		17 (65.4 %)	
>20 and <1000/mL	9 (42.9%)	3 (14.3%)		4 (15.4%)	0.12 ^
>1000/mL	2 (9.5%)	1 (4.8%)		3 (11.5%)	
No data	0	4 (19.1%)		2 (7.7%)	
**Immune Status, CD4^+^ Counts *****					
CD4 >500/mm^3^	10 (47.6%)	5 (23.8%)		9 (34.6%)	
CD4, 200–500/mm^3^	5 (23.8%)	8 (38.1%)		11 (42.3%)	0.35 ^
CD4 < 200/mm^3^	6 (23.6%)	5(23.8%)		4 (15.4%)	
No data	0	3 (14.3%)		2 (7.7%)	
**Antiretroviral Therapy (ART)**					
Yes	18 (85.7%)	16 (76.2%)		24 (92.3%)	0.10 ^
No	3 (14.3%)	1 (4.8%)		1 (3.9%)	
No data	0	4 (19.1%)		1 (3.9%)	

* Benign tissues were histopathologically confirmed to be non-lesional by a board-certified pathologist, they were obtained from patients undergoing surgery for anal fistulas, hemorrhoids, and from patients undergoing anal surgery for treatment of anal HSIL. ** HIV viral load: reportable range is 20 to 10,000,000 copies/mL. *** CD4 categories are <200, 200–500, >500 cells/mm^3^. ^#^ Kruskal–Wallis *p*-value ^ Fisher’s Exact *p*-value.

**Table 2 cancers-15-00660-t002:** Single and multiple HPV infections (HPV16, Non-16 HR HPV, LR) found in benign, HSIL, and SCC lesions in 68 PLWH and 47 HIV-negative individuals.

(a) Benign			
HPV Types	PLWH (N = 21)	HIV-Negative (N = 13)	*p*-Value
One HPV 16 only	0	1 (7.7%)	0.38
HPV 16 + Non-16 HR HPV *	0	0	
HPV 16 + LR **	1 (4.8%)	0	1.00
HPV 16 + Non-16 HR HPV + LR	1 (4.8%)	0	1.00
Non-16 HR HPV only	2 (9.5%)	0	0.51
More than one Non-16 HR HPV	0	0	
Non-16 HR HPV + LR	1 (4.8%)	0	1.00
One LR only	0	1 (7.7%)	0.38
More than one LR	5 (23.8%)	0	0.13
Type unknown	3 (14.3%)	2 (15.4%)	1.00
No HPV	8 (38.1%)	9 (69.2%)	0.16
HPV detected	13 (61.9%)	4 (30.8%)	0.16
Multiple HR *** types	1 (4.8%)	0	1.00
Multiple LR types	5 (23.8%)	0	0.13
Combination of HR + LR	3 (14.3%)	0	0.27
Multiple infection	8 (38.1%)	0	0.013
HPV 16 (single + multiple)	2 (9.5%)	1 (7.7%)	1.00
**(b) HSIL**			
**HPV Types**	**PLWH (N = 21)**	**HIV-Negative (N = 9)**	** *p* ** **-Value**
One HPV 16 only	9/21 (42.8%)	6/9 (66.7%)	0.43
HPV 16 + Non-16 HR HPV	3/21 (14.3%)	2/9 (22.2%)	0.62
HPV 16 + LR	2/21 (9.5%)	0	0.51
HPV 16 + Non-16 HR HPV + LR	0	0	
Non-16 HR HPV only	2/21 (9.5%)	0	0.51
More than one Non-16 HR HPV	0	0	
Non-16 HR HPV + LR	3/21(14.3%)	1/9 (11.11%)	1.00
One LR only	0	0	
More than one LR	0	0	
Type unknown	1/21 (4.7%)	0	1.00
No HPV	1/21 (4.7%)	0	1.00
HPV detected	20/21 (95.2%)	9/9 (100%)	1.00
Multiple HR types	3/21 (14.3%)	2/9 (22.2%)	0.62
Multiple LR types	0	0	
Combination of HR + LR	5/21 (23.8%)	1/9 (11.11%)	0.64
Multiple infection	8/21 (38%)	3/9 (33.3%)	1.00
HPV 16 (single + multiple)	14/21 (66.7%)	8/9 (88.9%)	0.37
**(c) SCC**			
**HPV Types**	**PLWH (N = 26)**	**HIV-Negative (N = 25)**	** *p* ** **-Value**
One HPV 16 only	11/26 (42.3%)	17/25 (68%)	0.093
HPV 16 + Non-16 HR HPV	3/26 (11.5%)	1/25 (4%)	0.61
HPV 16 + LR	4/26 (15.4 %)	0	0.11
HPV 16 + Non-16 HR HPV + LR	0	0	
Non-16 HR HPV only	4/26 (15.4 %)	0	0.11
More than one Non-16 HR HPV	2/26 (7.7 %)	1/25 (4%)	1.00
Non-16 HR HPV + LR	1/26 (3.8 %)	1/25 (4%)	1.00
One LR only	0	0	
More than one LR	0	1/25 (4%)	0.49
Type unknown	1/26 (3.8 %)	3/25 (12%)	0.35
No HPV	0	1/25 (4%)	0.49
HPV detected	26/26 (100%)	24/25 (96%)	0.49
Multiple HR types	5/26 (19.2%)	2/25 (8%)	0.42
Multiple LR types	0	1/25 (4%)	0.49
Combination of HR +LR	5/26 (19.2%)	1/25 (4%)	0.19
Multiple infection	10/26 (38.5%)	4/25 (16%)	0.12
HPV 16 (single + multiple)	18/26 (69.2%)	18/26 (72%)	1.00

* HR HPV types other than HPV 16 (Non-16 HR HPV); ** Low-risk HPV (LR); *** High-Risk HPV (HR).

**Table 3 cancers-15-00660-t003:** Associations of HPV 16 with anal disease, overall and stratified by presence of HIV or Non-16 HR HPVs.

Model	Stratum	N	Proportion of Samples with HSIL/SCC (95% CI);	RR [of (HSIL/SCC) vs. Benign] (95% CI); *p*-Value
	Overall	115	70.4 (62.6, 79.3)	
1	HPV 16−	54 ^#^	42.6 (30.2, 56.0)	2.23 (1.63 to 3.06); *p* < 0.0001
	HPV 16+	61	95.1 (85.8, 98.4)	
2	HPV 16−, HIV−	20	40.0 (23.4, 68.4)	
	HPV 16+, HIV−	27	96.3 (89.4, 100)	2.41 (1.40 to 4.14); *p*= 0.0015
	HPV 16−, HIV+	34	44.1 (30.2, 64.4)	1.10 (0.57 to 2.13); *p* = 0.77
	HPV 16+, HIV+	34	94.1 (86.5, 100)	2.35 (1.37 to 4.05); *p* = 0.002
3	HPV 16−, Non-16 HR HPV−	36 ^^^	22.2 (12.1, 40.9)	
	HPV 16+, Non-16 HR HPV−	51	96.1 (90.9, 100)	4.32 (2.34 to 7.99); *p* <0.0001
	HPV 16−, Non-16 HR HPV+	18	83.3 (67.8, 100)	3.75 (1.97 to 7.15); *p*< 0.0001
	HPV 16+, Non-16 HR HPV+	10	90.0 (73.2, 100)	4.05 (2.12 to 7.72); *p* <0.0001

^#^ Includes 10 participants with HPV type unknown (5 HIV−, 5 HIV+) and 19 with LR HPV type (10 HIV−, 9 HIV+). ^ Includes 29 participants with HPV type unknown or LR HPV type.

## Data Availability

The data presented in this study are available on request from the corresponding author.

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
