# Peer review of "HPV Type Distribution in Benign, High-Grade Squamous Intraepithelial Lesions and Squamous Cell Cancers of the Anus by HIV Status"

_cancers, 2023, doi:10.3390/cancers15030660_

Round 1

Reviewer 1 Report

Overall, a very good manuscript.

Table 1 – Looks messy in its current layout. If it could be simplified, is there no way that the age and sex data could compare the HIV+ves with the HIV-ves? Re age breakdowns, could you simplify to, lose <25 (no observations), then 25-49, 50-69, 70. Re HIV VL, could you simplify to undetectable (at the top), >20 - <1000, >1000.

Table 2 – also looks very messy. One point, the abbreviation ‘non-16 oncHPV’ is (a) rather unusual, and (b) takes up more space in a table. Using non-16 HR HPV, or other HR, or oHR, would be more intuitive and take up less space. The same phrase starts occurring in the text from line 211 onward. Same as in Table 1, could the line categories be simplified somehow?

Figure 1, top left graph. I can’t believe that one HSIL had HPV type 34! Some typo has crept in, presumably it should be HPV 31.

Discussion, L292-295. I am not convinced that systemic omics approaches will solve these problems, but laser capture microdissection and intra-lesional PCR certainly can.

Discussion, L301-305. It is not pragmatic nor economic to suggest altering the existing multi-valent HPV vaccine on this basis.

Discussion, L312-320. This is a very complex area, and although the hypothesis proposed is possible, no supporting mechanistic evidence is cited, and the proposed explanation seems unlikely to this reviewer. There are other possible explanations / hypotheses / evidence though. Founder effects are known to underlie the global distributions of individual HPV 16 clades, and also HPV 18. Wheeler et al (2013) doi: 10.1002/ijc.27608 conducted a population-based study of HPV genotypes in women in New Mexico and showed that after HPV16, the most common genotypes were HPV 39/51/59. She proposed that this may be due to founder effects related to Hispanic and Native American ancestry in that population. Keller et al (2018) doi: 10.1097/QAD.0000000000002005 showed that HPV16 prevalence is lower in African American compared with Caucasian women with HIV and cervical precancer, independent of immune status. The ethnic status of participants in this study is not given, but if a very much larger population as well as multiple control groups could be studied, it might be possible to begin to address the reasons behind the lower HPV16 prevalence in HIV+ve MSM. For now though, caution in making any conclusions is necessary, and I believe the text needs to reflect this.     

Reviewer 3 Report

Thank you to the authors for an interesting, pertinent and well presented piece of work. I have only minor suggestions to aid the reader:

1. Results 3.1.2 - not sure presenting risk differences for such small sample sizes here is helpful. Would suggest just presenting the numbers as they are

2. Table 2 - suggest three separate tables (a, b and c) or perhaps have benign, HSIL and SCC in a column on the left to help the reader understand the different groups.

3. 3.1.2 para 2 - typo in formatting 'negative samples only two' 

4. Discussion para 6 - not necessarily that increased non-HPV 16 in HIV population is immunological, may be confounded by exposure /  lifestyle factors (e.g. MSM) - 

Author Response

Dear Reviewer,

We would like to thank you for your objective evaluation and constructive comments to improve the manuscript.  Based on your suggestions and comments we have made changes to the manuscript. The revisions to the manuscript are marked by “Track changes”.

We have also added new references to the manuscript as per your recommendations. Please note, as per the directions of the Assistant Editor new references have been added at the end of the text; the references will be reordered/formatted by the Journal after resubmission.

 We have also addressed each of the comments in a detailed manner below.

Thank you for your consideration of this manuscript.

Sincerely,

Sona Chowdhury

[email protected]  

-----------------------------------------------------------------------

REVIEWER 3:

Comment 1: Results 3.1.2 - not sure presenting risk differences for such small sample sizes here is helpful. Would suggest just presenting the numbers as they are 

Response: We agree that given the small sample size that detecting the risk differences would not achieve statistical significance. However, given the large risk difference and clinically meaningful difference for the benign samples in 3.1.1, we would like to be consistent with the presentation of the results and provide the readership with the risk differences for the HSIL and SCC.

Comment 2: Table 2 - suggest three separate tables (a, b and c) or perhaps have benign, HSIL and SCC in a column on the left to help the reader understand the different groups.

Response: The editors do not want us to split Table 2 into sub-tables therefore we have combined it into one table

Comment 3: 3.1.2 para 2 - typo in formatting 'negative samples only two'

Done

Comment 4: Discussion para 6 - not necessarily that increased non-HPV 16 in HIV population is immunological, may be confounded by exposure / lifestyle factors (e.g. MSM) - 

Response: We have referenced the hypothesis regarding “HPV 16 possessing a better intrinsic ability to avoid the immune system”. We agree with your suggestion but would like to point out that there is evidence showing that HPV 16 is less affected by decrease in CD4+T cells counts in PLWHIV (Strickler et al, Lin et al) in comparison to other high-risk types. In comparison other HR types are more susceptible to immune surveillance and in absence of a functional immune system in PLWH these HR types profit to a greater degree than HPV 16.

“One of the hypothesis explaining this difference in HPV 16 frequency between these two groups (PLWH and HIV-negative) is that HPV 16 is less influenced by immunological deficiency and has better intrinsic ability to evade immune surveillance in immunocompetent individuals compared with non-16 oncHPV types”. 

Revised version: Line 342: One of the hypotheses explaining this difference in HPV 16 frequency between these two groups (PLWH and HIV-negative) is that HPV 16 is less influenced by immunological deficiency and has better intrinsic ability to evade immune surveillance in immunocompetent individuals compared with non-16 oncHPV types. Other than immunological deficiency, distribution of HPV types may vary depending upon other factors such as geographical distribution (Wheeler et al, Keller et al) and behavioral factors (Distribution of HPV Genotypes Differs Depending on Behavioural Factors among Young Women).

Reviewer 4 Report

Chowdhury et al. have performed a study on HPV-type distribution in anal biopsies of HIV-negative (n = 47) and HIV-positive (n = 68) patients. HPV16 was less common in premalignant and malignant lesions of HIV-positive patients compared to HIV-negative individuals. Infections with multiple HPV-types were found in tissues of HIV-positive patients and non-16 high-risk HPV-types were more frequent in anal cancers of this patient group. These findings are important for the development of anal cancer screening programs for people living with HIV.

I have some minor suggestions:

1. The number of samples analyzed should be mentioned in the abstract.

2. One or more references should be added after the sentences in lines 60-62 and 312-315, respectively.

3. Section 2.1

Please mention the time period in which the samples were collected.

4. Table 1: Column 2 (Benign*) seems to comprise both normal and benign lesions. Please show the results of normal and benign lesions in two separate columns.

5. As in Table 3, a column should be included in Table 2, showing whether the differences between PLWH and HIV-negative persons were statistically significant.

6. The limitations of the study should be mentioned in the discussion (e.g., small number of samples in the subgroups).

7. Please include and cite the following references

Wei et al. PMID: 34339628

Round 2

Reviewer 2 Report

Thanks for the revised manuscript. The point-by-point responses as well as the changes made addressed my concerns. I have a few minor comments left.

Lines 155-157: What do you mean “for ordered age data” in the parenthesis in Kruskal Wallis test? I think you should use Fisher’s test to assess association between the variables (age, sex,…) and disease status (benign, HSIL, SCC). The Kruskal-Wallis test is a nonparametric (distribution free) test, and is used when the assumptions of one-way ANOVA are not met. If you treat age as a categorical variable, Fisher’s exact test will be more appropriate. Although Kruskal Wallis test can be used for ordinal variable, it would be meaningful ordered all along its range and ranges are wide enough (e.g., Likert 1-10 scale).

Lines 159-160: I do not feel that you need to say “via the FREQ procedure of SAS”. If you want to say this, you should move “SAS v 9.4 was used to generate results” in the beginning of this section.

Lines 169: Shouldn’t SAS v.94 be SAS version 9.4?

Author Response

We are glad that the changes made addressed your concerns. Thank you. 

Lines 155-157: What do you mean “for ordered age data” in the parenthesis in Kruskal Wallis test?

We were highlighting the difference between the Kruskal Wallis test and the Fisher’s exact test.  We used the Kruskal Wallis test for the age variable since it is an ordinal categorical variable.

We removed the parenthesis to make it more clear why we also included the results from the Kruskal Wallis test. “Kruskal Wallis results were also reported comparisons for age, given that the categories are ordered.”

I think you should use Fisher’s test to assess association between the variables (age, sex,…) and disease status (benign, HSIL, SCC). The Kruskal-Wallis test is a nonparametric (distribution free) test, and is used when the assumptions of one-way ANOVA are not met. If you treat age as a categorical variable, Fisher’s exact test will be more appropriate. Although Kruskal Wallis test can be used for ordinal variable, it would be meaningful ordered all along its range and ranges are wide enough (e.g., Likert 1-10 scale).

The reason the Kruskal Wallis test was provided is because the age data are divided into mutually exclusive ordered discrete categories. The Fisher’s exact test does not account for the inherent ordering of the age categories, and we would arrive at the same answer in terms of the Fishers Exact p-value if we changed the ordering of the age-categories.  We included both results from the Fisher’s exact test and Kruskal Wallis test.  That is an interesting point about the ranges. Based on our review of the literature, the results should not be largely influenced by the data in terms of the ordering of the categories  as long as the categories are monotonically increasing. Based on your feedback, we included both results from the Fisher’s exact test and Kruskal Wallis test.

Lines 159-160: I do not feel that you need to say “via the FREQ procedure of SAS”. If you want to say this, you should move “SAS v 9.4 was used to generate results” in the beginning of this section.

We removed it. Thank you.

Lines 169: Shouldn’t SAS v.94 be SAS version 9.4?

That is correct, thank you for catching this typo.